

# Development of a novel nanoflow liquid chromatography-parallel reaction monitoring mass spectrometry-based method for quantification of angiotensin peptides in HUVEC cultures

Chuan He, Simiao Hu and Wanxing Zhou

The first affiliated hospital of Guangdong Pharmaceutical University, Guangdong, Peoples Republic of China
Guangdong Metabolic Disease Research Center of Integrated Chinese and Western Medicine, Guangdong, Peoples Republic of China

## ABSTRACT

**Background**. This study aimed to develop an analytical method using liquid chromatography tandem mass spectrometry (LC-MS/MS) for the determination of angiotensin (Ang) I, Ang (1-9), Ang II, Ang (1-7), Ang (1-5), Ang III, Ang IV in human umbilical vein endothelial cell (HUVEC) culture supernatant.

**Methods**. HUVEC culture supernatant was added with gradient concentrations (0.05–1,000 ng/ml) of standard solutions of the Ang peptides. These samples underwent C18 solid-phase extraction and separation using a preconcentration nano-liquid chromatography mass spectrometry system. The target peptides were detected by a Q Exactive quadrupole orbitrap high-resolution mass spectrometer in the parallel reaction monitoring mode. Ang converting enzyme (ACE) in HUVECs was silenced to examine Ang I metabolism.

**Results**. The limit of detection was 0.1 pg for Ang II and Ang III, and 0.5 pg for Ang (1-9), Ang (1-7), and Ang (1-5). The linear detection range was 0.1–2,000 pg (0.05–1,000 ng/ml) for Ang II and Ang III, and 0.5–2,000 pg (0.25–1,000 ng/ml) for Ang (1-9) and Ang (1-5). Intra-day and inter-day precisions (relative standard deviation) were <10%. Ang II, Ang III, Ang IV, and Ang (1-5) were positively correlated with ACE expression by HUVECs, while Ang I, Ang (1-7), and Ang (1-9) were negatively correlated.

**Conclusion**. The nanoflow liquid chromatography-parallel reaction monitoring mass spectrometry-based methodology established in this study can evaluate the Ang peptides simultaneously in HUVEC culture supernatant.

## INTRODUCTION

The angiotensin (Ang) family is part of the renin-angiotensin system (RAS) and is involved in cardiovascular homeostasis and health by regulating the blood pressure (*Basso & Terragno, 2001*). The family includes Ang I and its metabolites: Ang II, Ang III, Ang IV, Ang (1-9), Ang (1-7), and Ang (1-5) (Fig. S1). Ang I is hydrolyzed by the Ang converting

Corresponding author
Wanxing Zhou,
zhouwanx2015@sina.com

enzyme (ACE) to form Ang II (*Culver, Li & Siragy, 2017*). Ang II can induce intense arterial vasoconstriction and elevate blood pressure by activating the Ang type 1 receptor (AT1), induce endothelial dysfunction, and the pathological remodeling of cardiovascular tissue and aggravate water and sodium retention and cardiovascular remodeling by up-regulating aldosterone (*Touyz, 2003*). Ang I can also be hydrolyzed by the Ang converting enzyme 2 (ACE2) to form Ang (1-9), which is further hydrolyzed by ACE to form Ang (1-7) and Ang (1-5). Ang (1-9) can bind to the Ang type 2 receptor (AT2) and block Ang II-induced cardiac hypertrophy. Meanwhile, it can serve as a substrate for ACE, reducing Ang II production through competition with Ang I (*Buczko, Kramkowski & Mogielnicki, 2006*). Ang (1-7) modulates the nitric oxide release from endothelial cells through the Mas receptors (*Santos & Ferreira, 2007*) and inhibits sodium reabsorption in the kidney (*Castelo-Branco, Leite-Delova & De Mello-Aires, 2013*). In addition, Ang II can be hydrolyzed into Ang III, which has effects similar to Ang II (*Kemp et al., 2012*). Ang III can be further hydrolyzed to generate Ang IV, which can activate the nuclear factor (NF)-$\kappa$B to up-regulate the expression of interleukin (IL)-6 and tumor necrosis factor (TNF)-$\alpha$, which promote the proliferation and migration of vascular endothelial cells and smooth muscle cells, leading to cardiovascular remodeling (*Fyhrquist & Saijonmaa, 2008*). Hence, Ang metabolism is complex, and each member has specific biological functions, and they can interact with each other. Due to the large span and low content of Ang peptides in different body substrates (blood, urine, pleural effusion, and various tissues), great differences in concentrations of each Ang are observed (*Lortie et al., 2009*). Therefore, a method with high throughput, wide linear dynamic range, high sensitivity, and high reproducibility is needed. The traditional methods for detecting Ang peptides are immunological methods (*Chappell et al., 2012*; *Chappell, 2016*). In addition to the need for developing the antibodies, these methods have lower detection throughput, narrow linear dynamic range, and immune cross-reactivity among Ang metabolites because they share the same amino acid sequences (*Jankowski et al., 2011*).

Compared with immunological methods, liquid chromatography-mass spectrometry (LC-MS) directly analyzes the molecular structures, avoiding the need for antibody preparation. It has extremely high specificity and retains linearity over a wide concentration range (*Olkowicz, Chlopicki & Smolenski, 2017*; *Tamvakopoulos, 2007*). It can simultaneously detect multiple components in one sample. Liquid chromatography tandem mass spectrometry (LC-MS/MS) can be applied for the quantitative analysis of polypeptides. The new high-resolution mass spectrometer Q Exactive used in the present study can reach a resolution of 140,000 (*Eliuk & Makarov, 2015*). Parallel reaction monitoring (PRM) was developed on the basis of select reaction monitoring (SRM). PRM is based on a quadrupole high-resolution mass spectrometer platform that included the Q-Orbitrap device. Unlike the SRM that only select one specific fragment ion for detection and analysis in the secondary mass spectrometry, the PRM collects all the fragment ions after the fragmentation of the precursor ion, resulting in better reliability and specificity (*Malchow et al., 2017*; *Ronsein et al., 2015*; *Bourmaud, Gallien & Domon, 2016*). The first step of PRM is to use the quadrupole (Q1) to select the parent ion. The second step is to break the parent ion in the collision pool (Q2) to form the daughter ions. Finally, the Orbitrap is used to

replace Q3, and all the sub-ions are scanned in the high-resolution/high-quality precision (HR/AM) mode to complete PRM data acquisition (*Gallien et al., 2013*). Compared with the previous SRM/multiple reaction monitoring (MRM) used in quantitative Ang analysis, the PRM selected in this study has some advantages: (1) high-resolution sub-ion monitoring, quality accuracy up to the part-per-million level, and eliminating background interference in cell culture matrix to the greatest extent without loss of sensitivity; (2) two-stage mass spectrometry within a single full scan, which requires no prior determination of ion pairs and optimization of collision energy; only one or several fragment ions with the highest response can be selected during data processing to extract the chromatographic peaks of the ion pair for quantitative analysis, with a linear range of 5–6 orders of magnitude; and (3) both qualitative and quantitative analyses are performed simultaneously. The MS/MS full scan spectrum is used for qualitative analysis, and the optimal fragment ion can be used for quantitative analysis. High-throughput analysis of multiple Ang peptide data can save a lot of time in matching ion pairs and parameter optimization.

Human umbilical vein endothelial cells (HUVECs) have become an important model for studying vascular physiological functions and pathological remodeling (*Nachman & Jaffe, 2004*). Although previous studies reported the use of the SRM LC-MS/MS technique to detect the Ang peptides in blood, urine, and tissues (*Olkowicz, Chlopicki & Smolenski, 2017*; *Olkowicz et al., 2015*; *Ali et al., 2014*), the methods for detecting the peptides in cell culture supernatants were rarely reported (*Finoulst et al., 2011a*; *Finoulst et al., 2011b*; *Kay et al., 2018*). Culture media often contain substances that are not found in body fluids and tissues, and lowly-expressed peptides may be masked by more abundant ones, resulting in potential interference. In addition, there is a lack of studies for the mass spectrometry detection of Ang peptides using the PRM model.

We hypothesized that the PRM LC-MS/MS technique could be used to detect Ang peptides from HUVEC culture supernatants. Therefore, the present study aimed to establish a set of methods for the quantitative analysis of Ang peptides in HUVEC culture supernatants, more specifically Ang I, Ang (1-9), Ang II, Ang (1-7), Ang (1-5), Ang III, and Ang IV.

## MATERIAL AND METHODS

### Experimental design
This study was conducted in two phases. In the first phase, the LC-MS/MS protocol was optimized using culture supernatant. In the second phase, in order to examine whether the detection method was sensitive to changes in Ang metabolism, ACE was silenced in cells, and the Ang profile was measured.

### Reagents
The reagents are presented in the Supplementary Data.

### Cell lines
The human umbilical vein endothelial cells (HUVEC) were bought from ScienCell Research Laboratories (USA, Catalog Number: 8000, Lot Number: 14260). They were primary cells at passage 0.

## Optimization of the LC-MS/MS conditions and establishment of the PRM methodology

The pieces of equipment are presented in the Supplementary Data.

Ang I, Ang (1-7), Ang II, Ang (1-9), Ang (1-5), Ang III, and Ang IV standard solutions and internal standard solutions were prepared in 0.1% formic acid water solution at 1 μg/ml of the Ang peptide. The solutions (2 μl) were loaded at 5 μl/min. Mobile phase A was 0.1% formic acid water solution. Mobile phase B was 0.1% formic acid acetonitrile solution. The elution gradient was 2% B at 0 min, 4% B at 2 min, 22% B at 47 min, 35% B at 57 min, 90% B at 62 min, and 92% B at 70 min. The flow rate was 250 nl/min.

The nano-ESI was combined with a positive ion mode. Spray voltage was 2.25 kV. The capillary temperature was 320 °C. The S-lens was 50%. The full scan+ ddMS$^2$ mode was used. Resolution setting were full scan 70,000 and ddMS$^2$ 17,500. The precursor ion scan range was m/z 250–1800. The collision energy was 27% HCD (high energy collision dissociation). The acceptable automatic gain control (AGC) target in full scan mode was setted at 3e6, and in ddMS$^2$ mode was setted at 5e4. The maximum injection time (maximum IT) in full scan was setted at 60 ms, and in ddMS$^2$ mode was setted at 80 ms. The Skyline Ang spectra library was constructed based on the amino acid sequence of the peptides. The detected mass spectral data were searched through a database, and the best precursor ion of the target peptide was selected for establishing the method.

The precursor ion mass-to-charge ratio (m/z) of the target peptide was added to the inclusion list to establish a mass spectrometry acquisition method. PRM data of the sample was acquired using the PRM method and to determine the final PRM method for sample data collection. The resolution was MS: 70,000. The MS/MS ratio was 35,000. The precursor ion scan range was m/z 300–1500. The AGC target was 3e6 for MS and 2e5 for MS/MS. The maximum IT was 60 ms for MS and 100 ms for MS/MS. The fixed first mass was 100 m/z. The isolated window was 1.6 m/z. The MS information about the Ang family is presented in Table S4.

## HUVEC culture and preparation of cell supernatant matrix solution

HUVECs were inoculated in culture flasks and grown to 70% confluence. The original culture medium was discarded. The cells were washed three times with phosphate buffer saline (PBS). Phenol red-free endothelial cell basal medium (ECM-prf) (5 ml) and 1% endothelial cell growth supplements (ECGS) were added, and the cells were cultured at 37 °C with 5% $CO_2$ for 24 h. Culture supernatants were harvested and centrifuged at 2000 g, 4 °C, for 15 min. A mass spectrometry-specific protease inhibitor mixture (Beyotime Biotechnology, China) and 50 mM ethylene diamine tetraacetic acid (EDTA) solution were added at 1:50, and 576 mM 2-mercaptoethanol (Sigma, St Louis, MO, USA) was added at 1:40. The final cell supernatant solution was stored at −80 °C.

## Development and validation of the methodology
### Evaluation of linear dynamic range and sensitivity

Trifluoroacetic acid (TFA) (2.5%) was added to the HUVEC culture supernatant solution to achieve a final TFA concentration of 0.2%. After centrifugation at 12,000 g, 4 °C, for 30 min, the supernatant was mixed with the standard peptide solution to prepare a gradient

concentration of 10 samples: 1,000, 500, 100, 20, 10, 5, 1, 0.5, 0.25, and 0.05 ng/ml. The internal standard $Val_5$-Ang II was 100 ng/ml.

The Sep-Pak Vac solid-phase extraction (SPE) C18 (1 cc, 100 mg, Waters, Milford, MA, USA) was treated with pure acetonitrile, three times, 1 ml/time, and pre-equilibrated with 0.1% TFA, three times, 1 ml/time. The sample (100 μl) was absorbed and diluted with four volumes of 0.1% TFA. After dilution, the sample was loaded on the equilibrated SPE cartridge at 3 ml/min. The sample passed through the bed of the cartridge that was collected and loaded twice, and then washed three times with 5% methanol, 1 ml/time. The cartridge underwent continuous vacuum drying for 5 min to allow the bed to be completely dry. Then, 80% acetonitrile/0.1% TFA was used for three elutions, with 500 ml each. The eluent was collected in centrifuge tubes. The 'Concentrator plus' vacuum centrifugation concentrator (Eppendorf, Hamburg, Germany) was used for vacuum centrifugation and swab-off of the samples.

The 0.1% formic acid/2% acetonitrile solution (100 μl) was used for re-dissolution, and 2 μl of the sample was loaded. The ratio and the theoretical content were used to calculate the correlation coefficient, slope, and intercept of the linear regression equation, and the standard curve was plotted. The lower limit of quantitation (LLOQ) was a signal-to-noise ratio (S/N) $\geq 10$.

### *Evaluation of precision*

The mixed Ang peptide solution at a concentration of 10 ng/ml was selected for quality control (QC). For intra-day precision evaluation, QC samples were repeatedly tested five times on the same day. For inter-day precision evaluation, QC samples were pre-processed in the same way and repeatedly tested for 3 days. The detection precision was assessed by the intra-day and inter-day relative standard deviation (RSD). RSD = (standard deviation of Ang peptide quantitative fragment ion peak area/average value of Ang peptide quantitative fragment ion peak area)*100%.

### *Sample stability*

The minimum operating temperature of the liquid sample injector was 7 °C. All samples were stored at −80 °C and prepared in batches of groups of three samples before the test. The test time of each sample was 80 min, and the total operation time for one group was 4 h. QC samples were tested on the machine after pretreatment and repeated six times. Percentage of sample stability = (peak area of quantitative fragment ions for Ang in loading sample/peak area of quantitative fragment ions for Ang in the first loading sample) *100%. A percentage of 85%–115% was considered stable.

## Effect of pretreatment on sample detection
### *Effect of solid-phase extraction column loading frequency on angiotensin detection*

The number of repeated loading on the SPE cartridge was evaluated using two, three, or four times. The QC sample solution (100 μl) was diluted with 400 μl of 0.1% TFA. The column solution was collected after the sample was passed through the SPE cartridge. The

samples were detected to compare the differences in the detection value of Ang peptides after different numbers of SPE treatments.

### Effect of ultrafiltration on angiotensin peptide analysis

In the SPE group, 100 µl of the QC sample solution was diluted with 400 µl of 0.1% TFA. The sample was loaded on the bed of the cartridge three times, according to the above method. For the ultrafiltration centrifugation + SPE group, ultrafiltration centrifugation was first performed to remove the proteins; 100 µl of the QC sample solution was added to the inner cannula of an ultrafiltration tube (MWCO 10 kDa, PES, Pall Life Sciences, Ann Arbor, MI, USA) and centrifuged at 4 °C, 12,000 rpm for 10 min. Then, 200 µl of the 0.1% TFA solution was added to the inner cannula for rinsing. It was centrifuged and then rinsed and centrifuged. The liquid in the outer cannula was collected. Then, the above SPE operation was performed again.

The samples in the two groups were tested after SPE to compare the difference in the detection value of Ang peptides in the two groups and evaluate the effect of ultrafiltration on the Ang peptide analysis in samples.

## Effect of different ACE-expressing HUVECs on Ang I metabolism

The RNA interference technique was used to construct model cell lines with lentivirus-based shRNA interference sequences to stably silence ACE expression in HUVECs. The construction method and results are shown in the Supplementary Material. The HUVEC model group with a negative sequence introduction and normal ACE expression was the PLKO.1 group. HUVECs with the ACE-shRNA interference sequence introduction and stable silencing of ACE gene expression were the interference group. It was divided into the ACE-shRNA-1 and ACE-shRNA-2 group according to 99% and 73% downregulation of ACE mRNA levels. See the Supplementary Data for the methods regarding the detection of Ang I and its metabolites.

## Data analysis

Skyline was used to import the mass spectrometric data files. The three strongest precursor ion (M, M+H, and M+2H) peaks for each peptide were extracted. The ion match tolerability was 0.01 m/z, satisfying idopt $\geq$ 0.9. The eight strongest y, b fragment ion peaks for each peptide were extracted. One of the quantitative fragment ion, complex ion, precursor ion match information, and retention time with high specificity and intensity was selected for qualitative analysis of Ang metabonomics. The relative quantitative analysis was performed using the ratio of Ang peptide fragment ion area to the internal standard fragment ion area. The standard curve method was used for absolute quantitative analysis of Ang. The $t$-test was used to analyze the differences between the two groups. Differences among the three groups were compared using one-way ANOVA. $P < 0.05$ was considered statistically significant. The CORREL function in MS Excel (Microsoft, Redmond, WA, USA) was used to analyze the correlations between the two sets of data and plot the correlation matrix.

## RESULTS

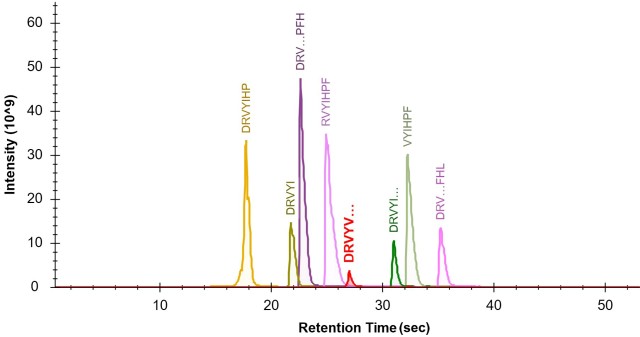

**Figure 1** **Optimization of the LCMS/MS parameters.** Chromatogram of the seven target peptides and internal standard peptide by LC-MS/MS. The seven target peptides and the internal standard peptide could complete the column elution with acetonitrile concentrations of 10%–16%. In order to separate the elution peaks and not affect the peak shape, the separation stages were set using a gradient from 4% B at 2 min to 22% B at 47 min. It can be seen that the peaks of each target peptide and internal standard peptide were smooth and effectively separated.

### The LC-MS/MS parameters were optimized, and the PRM detection model was established

The seven target peptides and the internal standard peptide could complete the column elution with acetonitrile concentrations of 10%–16%. In order to separate the elution peaks and not affect the peak shape, the separation stages were set using 4% B at 2 min to 22% B at 47 min. The peaks of the target peptide and internal standard peptide were smooth and separated (Fig. 1). The optimal sample dilution and loading volume were determined according to the different peak concentrations of the sample as 100 μl for re-dissolution and loading volume of 2 μl.

Mass spectrometry parameters were optimized by tuning the capillary voltage, collision energy, and ion dwell time. At a capillary voltage of 2.25 kV, the (M+3H) precursor ions of Ang I and Ang (1-9) were mainly the charged precursor ions, and the remaining of the target peptides were mainly the $(M+2H)^{2+}$ precursor ions. A target peptide list of the PRM method was established based on these results, and the ion dwell time was adjusted according to the number of precursor ions. Using the peptide map library pre-built using Skyline, the fragment ions in the PRM scan data were extracted (Fig. 2). The fracture law and abundance of fragment ions were matched with the spectrum library. The mass spectral response was up to $10^6$ orders of magnitude. The signal-to-noise ratio was high, and the mass spectral peak shape was smooth.

### The standard curves were built, and the detection limits were determined

The linear correlation coefficients of the standard curves were analyzed using least-squares regressions and were all ≥0.99 (Fig. S2), suggesting high detection accuracy. The results of the slope, intercept, linear range, and the LLOQ of the LC-MS/MS linear detection (Table S5) showed that the linear dynamic ranges of most Ang peptides reached 2–4 orders of magnitude, and the LLOQ reached among 0.05–0.25ng/ml. This method had a wide linear

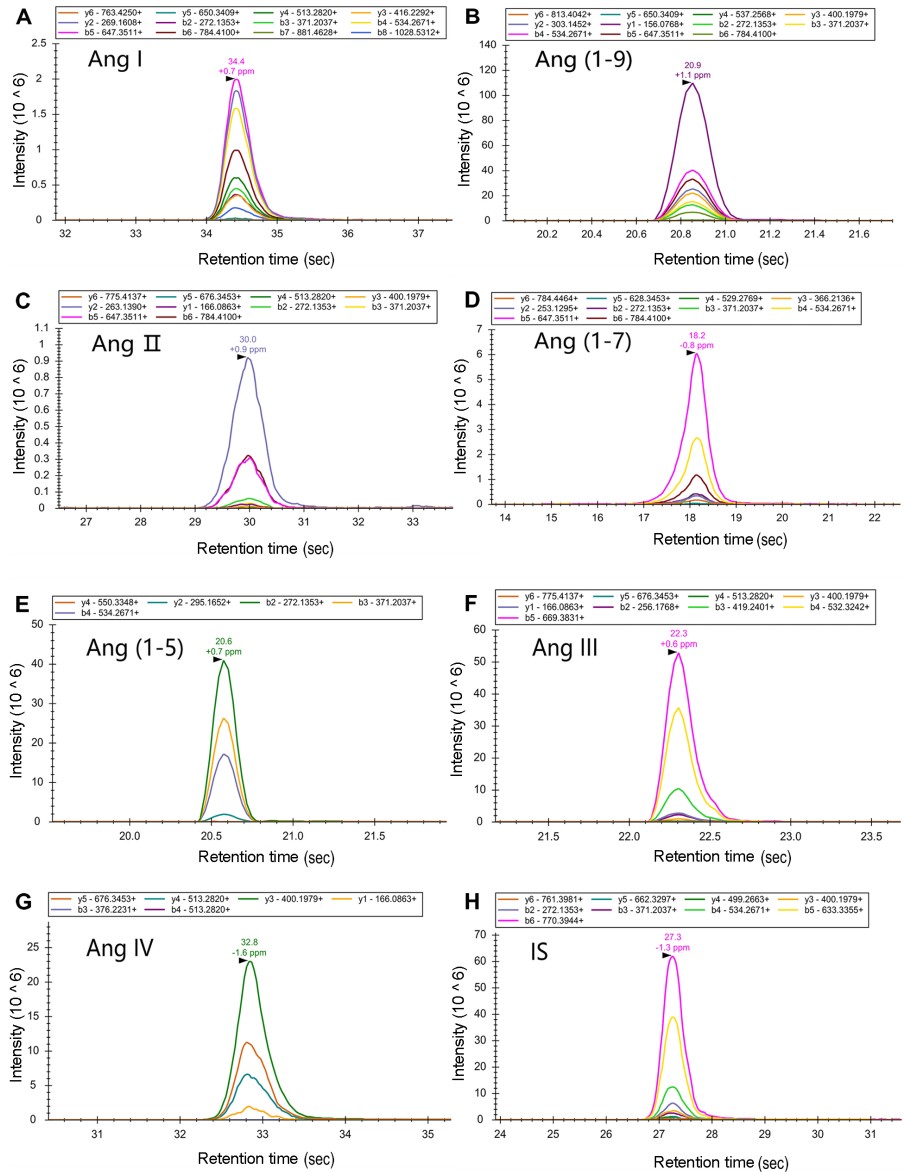

**Figure 2 Establishment of the PRM detection model.** The ion dwell time for Ang I, Ang (1-9), Ang II, and Ang (1-7) was adjusted according to the number of precursor ions. Using the peptide map library pre-built using Skyline, the fragment ions in the PRM scan data were extracted. The fracture law and abundance of fragment ions were matched with the spectrum library. The mass spectral response was up to $10^6$ orders of magnitude. The signal-to-noise ratio was high, and the mass spectral peak shape was smooth.

dynamic range and high sensitivity and could satisfy the detection of multiple Ang peptides with large differences in concentration.

## The precision was determined

The intra-day precision of the seven Ang peptides in the QC sample was RSD<10%, and the inter-day precision was also RSD <10% (Table S6). All errors were within the allowable

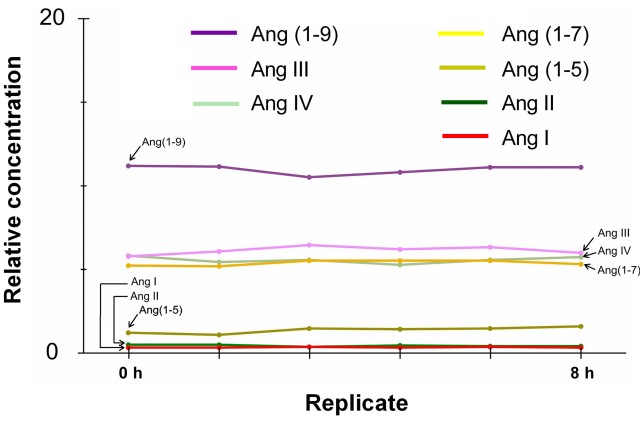

**Figure 3** **Determination of the sample stability over time.** Stability (% of changes in concentration) of the seven angiotensin peptides kept in the injector at 7 °C over 8 h.

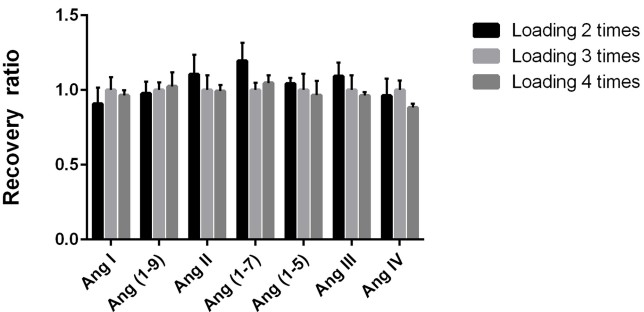

**Figure 4** **Determination of the number of sample loading on Ang detection.** Evaluation of the number of sample passages through the reverse C18 solid-phase extraction column packing on the recovery rate of angiotensin peptides. We used three loadings on the cartridge as control and evaluated the other numbers of loading times on the recovery rate of Ang by comparing the peak area ratio of the same target peptide quantitative fragment ion. The control group used three passages.

range, indicating that this method had high repeatability and optimal stability in detecting the Ang metabolites in the cell culture supernatant.

## The sample stability was determined

The Ang peptides were stable at 7 °C in the liquid sample injector for 6 h, and the concentration change was <15% (Fig. 3).

## The loading times did not affect sample recovery

We used three loadings on the cartridge as control and evaluated the other numbers of loading times on the recovery rate of Ang by comparing the peak area ratio of the same target peptide quantitative fragment ion. There were no significant differences in the recovery rate among different loading times (all CV <11.9%, Fig. 4), suggesting that the recovery rate was stable when the sample was loaded on the cartridge for two, three, or four times.
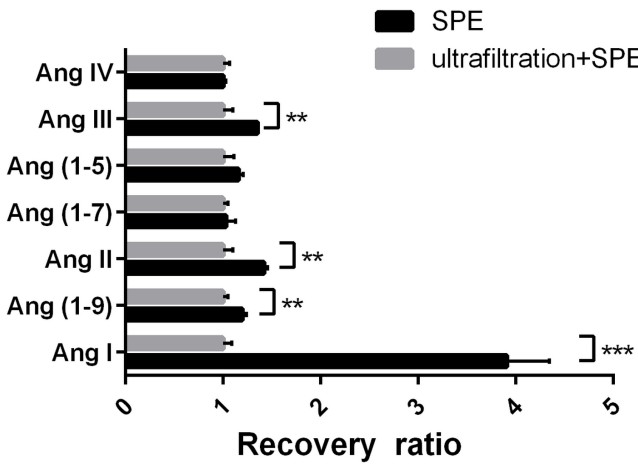

**Figure 5** **Determination of the impact of ultrafiltration on sample recovery.** Recovery rate of angiotensin peptides after ultrafiltration and solid-phase extraction compared with solid-phase extraction alone. **$P < 0.01$. ***$P < 0.001$.

### Ultrafiltration reduced sample recovery

The peak area of the Ang fragment ion in the ultrafiltration and SPE group was used as a normalization factor. The recovery rate of Ang I in the SPE group was significantly higher (3.91 times) than in the ultrafiltration and SPE group. The recovery ratios of Ang (1-9), Ang II, and Ang III in the SPE group were higher than in the ultrafiltration and SPE group (1.20, 1.42, and 1.35 times high, respectively). There were no significant differences in the recovery rates of other target peptides between the two groups (CV <10%, Fig. 5). This indicated that ultrafiltration would significantly reduce the recovery rate of Ang I, as well as Ang II, Ang III, and Ang (1-9).

### ACE protein expression was successfully silenced in HUVECs

The mRNA results show that the two interfering sequences caused the ACE mRNA to be down-regulated by 73% and 99% (Table S3, Fig. S3). Western blotting showed that the expression of the ACE protein in the ACE-shRNA-1 and ACE-shRNA-2 groups was significantly reduced (95.6% and 78.2%, respectively) compared with the PLKO.1 group (Fig. 6).

### ACE silencing induces differences in the Ang metabolic profile that can be detected by LC-MS/MS

To verify the practicability of the Ang assay established in this study, we added Ang I at a final concentration of 1 μM to three groups of HUVECs with different ACE expression levels. After incubation for 1 h, Ang I and its major metabolites (Ang II, Ang (1-7) and Ang (1-5), Ang (1-9), Ang III, and Ang IV) were detected by the LC-MS/MS method established in this study. The Ang II, Ang III, Ang IV, and Ang (1-5) levels in the ACE-shRNA-1 and ACE-shRNA-2 groups were significantly reduced compared with that in the PLKO.1 group. The decrease in the ACE-shRNA-1 group was more obvious than in the ACE-shRNA-2 group. Ang (1-9) and Ang (1-7) levels were significantly elevated, and the increase of Ang

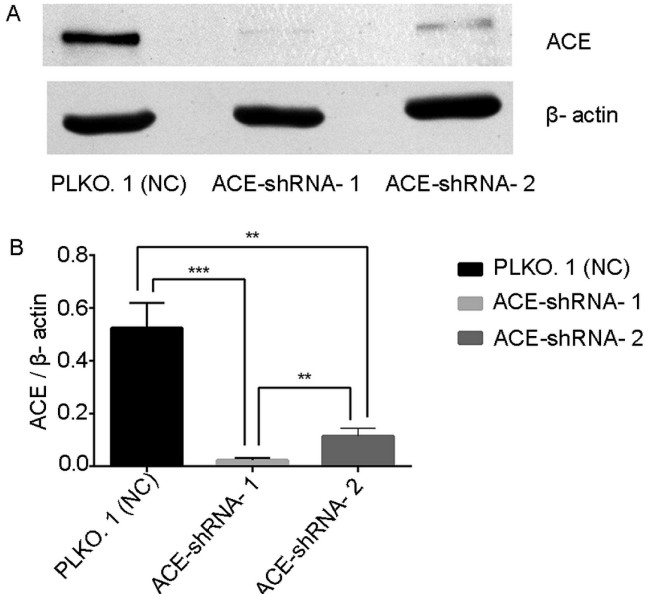

**Figure 6** **ACE protein expression was successfully silenced in HUVECs.** Western blotting showing that the expression of the ACE protein in the ACE-shRNA-1 and ACE-shRNA-2 groups was significantly reduced (95.6% and 78.2%, respectively) compared with the PLKO.1 group. $**P < 0.01$. $***P < 0.001$.

(1-9) in the ACE-shRNA-2 group was more obvious than in the ACE-shRNA-1 group. Ang (1-7) had an increasing trend. The Ang I concentrations in the ACE-shRNA-1 and ACE-shRNA-2 groups were significantly higher than in the PLKO.1 group (Fig. 7). Ang I, Ang (1-9), and Ang (1-7) concentrations in the culture supernatant of HUVECs were significantly and negatively correlated with the expression of ACE. The Ang II, Ang III, Ang IV, and Ang (1-5) concentrations in the culture supernatants showed significant positive correlations with ACE protein expression (Fig. S4).

## DISCUSSION

In the present study, the slope, intercept, linear range, and the LLOQ of the LC-MS/MS linear detection showed that the linear dynamic ranges of most Ang peptides reached 2-4 orders of magnitude, and the LLOQ reached the level of the femtogram. Previous studies using LC-MS/MS to detect Ang peptides in biological samples also examined the linear dynamic range and quantitation limits of the method. *Cui, Nithipatikom & Campbell (2007)* detected Ang II, Ang III, Ang IV, and Ang (1-7) in a bovine endothelial cell culture system; the linear dynamic range was 50–1,250 pg on the column, concentration, and injection volume unknown, one order of magnitude, and the limit of detection was 25 pg on the column. *Suski et al. (2014)* established a method for the detection of Ang II and Ang (1-7) in a rat smooth muscle cell culture system; the linear dynamic range was 2.5–250 ng/ml (about 5–500 pg on the column, linear coefficient unknown), with two orders of magnitude. *Bujak-Gizycka et al. (2007)* detected the metabolites of Ang I in an EA.hy926 cell culture system; the dynamic linear range was 20 pM–100 nM (about 0.02 ng/ml-100ng/ml,

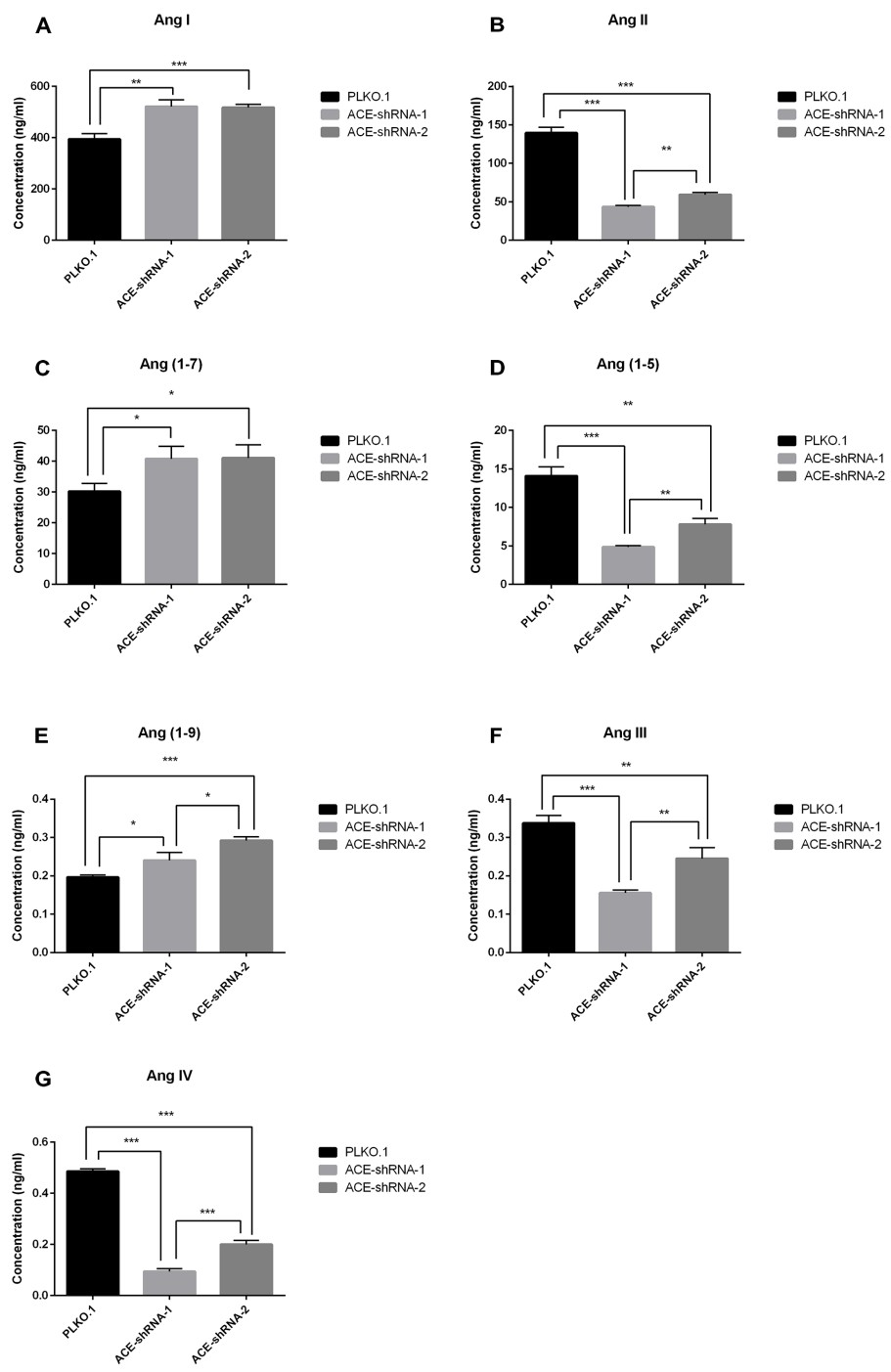

**Figure 7 ACE silencing induces differences in the Ang metabolic profile that can be detected by LC-MS/MS.** To verify the practicability of the Ang assay established in this study, Ang I was added at a final concentration of 1 μM to three groups of HUVECs displaying different levels of ACE expression. After incubation for 1 h, Ang I and its major metabolites (Ang II, Ang (1-7) and Ang (1-5), Ang (1-9), Ang III and Ang IV) were detected by the nano-LC-MS/MS method established in this study. *$P < 0.05$. **$P < 0.01$. *** $P < 0.001$.

0.1–500 pg on the column, MW was set in 1 kD, linear coefficient unknown), with three orders of magnitude.

In the present study, the Ang peptides were stable at 7 °C in the liquid sample injector for 6 h, and the concentration change was <15%. Previous studies showed that the optimal Ang storage conditions in serum samples were −80 °C for 1 month, repeated freeze-thaw for three times, and room temperature for 3 h (*Olkowicz, Chlopicki & Smolenski, 2017*; *Olkowicz et al., 2015*). Ang I and metabolites can remain stable for 12 h at 4 °C in an autosampler with a loss of <15%. To increase the stability of the sample and reduce the loss of Ang peptides, the following points may be noted. First, a low-temperature operation is required during collection, and various protease inhibitors should be added in time to fully inhibit the proteases involved in Ang metabolism (*Olkowicz, Chlopicki & Smolenski, 2017*; *Olkowicz et al., 2015*). In addition, sample pretreatment should be completed as fast as possible. The samples that cannot be promptly pre-treated should be quick-frozen in liquid nitrogen and stored at −80 °C. It should be rapidly thawed at room temperature before use to avoid the recovery of protease activity caused by slow warming. Second, the sample should be analyzed as soon as possible after the completion of the pretreatment process. It has been reported that Ang peptides are lost during long-term storage regardless of the conditions (*Lortie et al., 2009*; *Olkowicz et al., 2015*; *Ali et al., 2014*). It is related to the metabolic activities of enzymes in biological matrixes and material adsorption in storage containers. For some samples that cannot be immediately tested, they can be vacuum-dried and stored at −80 °C. Third, the sample bottles used for analysis were surface-silicified or polypropylene micro-injection bottles pre-coated with a saturated peptide mixture were used, avoiding the use of conventional glass insert-pipes. Small volumes dissolution was performed before loading on the machine to reduce the absorption of Ang peptides by the sample containers (*Lortie et al., 2009*; *Stejskal, Potesil & Zdrahal, 2013*).

Ultrafiltration is a commonly used method for sample pretreatment before LC-MS to separate proteins and small molecules. It is expected to increase the purity of the sample. The exfoliated cells, cell debris, other particulate-like substances, and some large proteins in the cell supernatant can be removed during ultrafiltration, leaving small peptides. In addition, proteases can be removed, increasing sample stability. On the other hand, ultrafiltration may also cause an unexpected loss of Ang peptides. Some authors have used ultrafiltration to purify serum and used LC-MS to detect Ang I and Ang II (*Lortie et al., 2009*; *Chappell et al., 2012*), but they did not evaluate the impact of ultrafiltration on the recovery rate of Ang in cell culture supernatants. The reduced recovery rate of Ang peptides could be related to the electrostatic interactions between the basic peptide and the negatively charged proteins and the hydrophobic interaction between the hydrophobic peptides and the materials of the ultrafiltration tube and filter membrane. The ultrafiltration tube lining used in this study was made of polypropylene, and the ultrafiltration membrane was made of polyethersulfone. It has been reported that polyethersulfone can absorb a large number of polypeptide molecules with nonpolar side chains. As a result, polypeptide molecules were trapped, and the recovery rate was reduced (*Hu & Kamberi, 2009*). Ang I, Ang (1-9), Ang II, and Ang III have phenylalanine at their carboxyl terminus, with strong hydrophobicity. Adding a certain proportion of surfactant or nonpolar solvent, or using

acetonitrile to reduce the polarity of the cell supernatant in the sample solution, would reduce the adsorption of Ang peptide by the ultrafiltration materials (*Hu & Kamberi, 2009*). Selecting an ultrafiltration membrane and ultrafiltration tube with low adsorption materials (such as cellulose acetate) may also improve its recovery rate. An acid or a base could be added to the collected samples so that the proteins and polypeptides have the same charges in order to reduce electrostatic adsorption (*Che et al., 2010*).

The eluted sample after SPE was finally dried by using a vacuum centrifugal concentrator to remove the elution solvent. It reduced the contamination and loss among samples compared with the nitrogen blowing method and was relatively simple and economical compared with the freeze-drying method. In the view of the experiences by others and our study, the following points may be the key to improve the pretreatment of the samples for the detection of Ang peptides in cell supernatants using LC-MS. First, the cell culture medium was routinely added with phenol red as an acid–base indicator. In our previous work, we found that phenol red was a high-retention material in the chromatographic system and would severely interfere with the purification and separation of Ang. Therefore, a phenol red-free medium is suggested for the use of LC-MS. Secondly, Ang peptides have biological activity and are sensitive to microorganisms. Therefore, the collection and pretreatment of samples should be conducted with strictly aseptic conditions. Thirdly, the addition of TFA to the sample for acidification can reduce the adsorption between basic Ang peptides and negatively charged macromolecular proteins in neutral solutions, promoting protein denaturation and precipitation (*Chertov et al., 2004*). On the other hand, acting as a strong ion pair reagent, TFA can improve the retention of Ang peptides in the C18 chromatographic columns. Fourthly, there were a lot of exogenously added amino acids and peptides in the cell culture medium, which would compete with the Ang peptides for the adsorption of the C18 column. The overloading phenomenon occurred when we used the micro SPE tips (Ziptip, Millipore, USA) to process the 100-$\mu$l sample in our previous study. Therefore, the particularities of cell culture supernatant should be fully considered, and a high-load SPE cartridge should be selected.

ACE is a key metabolic enzyme in the RAS and is involved in multiple metabolic pathways of Ang (*Culver, Li & Siragy, 2017*). Ang II is generated by ACE from Ang I, and Ang III and Ang IV are downstream metabolites of Ang II. Ang (1-5) is generated by ACE from Ang (1-7). Ang I, Ang (1- 9), and Ang (1-7) are regulated by both ACE and ACE2 (Fig. S1). Therefore, as supported by the known metabolism of Ang (*Streatfeild-James et al., 1998*; *Morris, Sanghavi & Kahwaji, 2020*; *Yugandhar & Clark, 2013*; *Chai et al., 2004*; *Jackman et al., 2002*; *Yu et al., 2016*), Ang II, Ang III, Ang IV, and Ang (1-5) were significantly reduced in culture supernatants when ACE expression was inhibited. The down-regulation of ACE expression inhibited the Ang I-ACE-Ang II pathway, and the Ang I concentration was higher than in the normal ACE expression group. In this study, there was a significant difference in ACE expression levels between the ACE-shRNA-1 group and the ACE-shRNA-2 group, but the Ang I levels were not significantly different between the two groups. This may be related to an increase in ACE2 expression after down-regulation of ACE (*Zhong et al., 2010*). In addition, the activity of the Ang I-NEP/MP/E-Ang (1-7) pathway could be potentiated. The upregulation of these two pathways can increase Ang

I catabolism. Therefore, it can explain the similar Ang I levels in different groups with different degrees of ACE inhibition. Down-regulation of the Ang (1-7)-ACE-Ang (1-5) pathway, the weakening of Ang (1-7) catabolism also increased the Ang (1-7) levels and decreased the Ang (1-5) levels. It can be seen that the down-regulation of ACE not only interfered with one metabolic pathway but also affected the entire Ang metabolic network.

Due to the low-quality resolution, it is difficult for MRM to remove the interference of complex matrix background effectively, and MRM is prone to false positive (*Sherman et al., 2009*; *Abbatiello et al., 2010*). On the other hand, with the increasing requirement of analytical flux, multiple ion pairs may need to be monitored in one analysis, while the limitation of MRM speed and sensitivity makes the number of ion pairs that can be monitored at the same time limited (*Kiyonami et al., 2011*). In addition, the optimization of ion pair, collision energy, and other conditions is time-consuming and laborious, which is difficult to meet the need for multiple product monitoring for metabolic network analysis (*Kiyonami et al., 2011*; *Cima et al., 2011*). Unlike the MRM approach, the quantitative fragment ions were not screened before the detection in the present study. We only first performed qualitative analysis using all the fragment ions with the help of the Skyline mass spectrometry analysis software in the later stage. During the quantitative analysis, we selected a fragment ion with a high signal as the quantitative fragment ion, which saved a great deal of method development time compared with the MRM method, while the sensitivity reached 0.1–0.5 pg on the column. It demonstrated the superior qualitative and quantitative capabilities of preconcentration nano-liquid chromatography coupled with quadrupole electrostatic field orbitrap mass spectrometry. Secondly, in this study, the biological matrix was the supernatant from HUVECs cultured in the endothelial cell culture medium, which could be used continuously for cell culture. Therefore, the method established has a more extensive application value.

In this study, a homologous internal standard was selected, in which valine was used to replace the 5th isoleucine of Ang II, with the remaining amino acid sequence being unchanged. The internal isotopic standard is the gold standard for absolute quantification using LC-MS, but the isotopic internal standard was not used in this study based on two considerations. First, this study does not rely on the concentration of internal isotopic standard for the absolute quantification of samples, and the internal standard is only used as the homogenization standard for pretreatment to correct the errors caused by pretreatment. Second, homologous internal indices were also used in the literature (*Olkowicz, Chlopicki & Smolenski, 2017*; *Olkowicz et al., 2015*; *Tyrankiewicz et al., 2018*). Undoubtedly, a stable isotopic internal standard is the most accurate choice, but its preparation cost is high. Due to limited funding, the homologous internal standard was used.

## CONCLUSIONS

The nano-liquid chromatography mass spectrometry with parallel reaction monitoring established in this study can simultaneously and accurately detect multiple Ang peptides in the culture supernatant of HUVECs. This method has the advantages of high sensitivity, wide linear dynamic range, large throughput, and high efficiency. It is an advanced and

reliable method for the study of cellular metabonomics of Ang. This method may be an innovative method for assessing the effect of RNA interference or therapeutic effects of ACE inhibitors, providing new ideas and targets for the role of RAS in the research and treatment of hypertension and cardiovascular remodeling.

### Funding
This work was supported by the science and technology plan of Guangdong provincial science and technology department (grant number 2017ZC0204). The funders had no role in study design, data collection and analysis, decision to publish, or preparation of the manuscript.

### Grant Disclosures
The following grant information was disclosed by the authors:
Guangdong provincial science and technology department: 2017ZC0204.

### Competing Interests
The authors declare there are no competing interests.

### Author Contributions
- Chuan He conceived and designed the experiments, performed the experiments, prepared figures and/or tables, and approved the final draft.
- Simiao Hu analyzed the data, prepared figures and/or tables, and approved the final draft.
- Wanxing Zhou performed the experiments, analyzed the data, prepared figures and/or tables, authored or reviewed drafts of the paper, and approved the final draft.

### Data Availability
Data are uploaded as Skyline files.
Skyline (https://skyline.ms/project/home/software/skyline/begin.view) was used to import the mass spectrometric data files.

### Supplemental Information
Supplemental information for this article can be found online at http://dx.doi.org/10.7717/peerj.9941#supplemental-information.

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
