# Peer review of "Development of a novel nanoflow liquid chromatography-parallel reaction monitoring mass spectrometry-based method for quantification of angiotensin peptides in HUVEC cultures"

_PeerJ, doi:10.7717/peerj.9941_

## Round 0.1 · original submission · Major Revisions

This is an invited resubmission of a manuscript originally rejected as submission #41659.

The authors should address mainly some technical issues, as indicated by the reviewers, and correct/improve figures.

·

Basic reporting

This article is written in correct English. It is conform to professional standards os courtesy and expression.
This article should be develop some literature approaches notably in MRM approaches which are the most used actually.
The structure of the article should conform to an acceptable format but some figures must be changed.

Experimental design

This article aimed to develop an analytical method using liquid chromatography tandem mass spectrometry (LC-MS/MS) for the determination of angiotensin peptides (Ang I, Ang (1-9), Ang II, Ang (1-7), Ang (1-5), Ang III, Ang IV) in human umbilical vein endothelial cell (HUVEC) culture supernatant. However some studies have been already developped this type of approaches. The originality lies in the use of PRM mode and nano-HPLC. Nevertheless the authors do not speak of the MRM approaches developed previously which are also sensitive.
Why not use a nano-LC-MS/MS

Validity of the findings

Some problems in this study :

- Only one peptide is used to identify and quantify
- We don't see any extraction percentage ? matrix effect ? (which is very important)
- The authors don't use radiolabelled standards (only one why ?). Indeed, ACE silencing is partial and it's was better to add for example a radiolabelled AngI.

- In Figure 1 : optimization of LC-MS/MS parameters seams not good. We do not see well on the figure but it's seams FWHM was too large (>30s). With this kind of nano-HPLC system it's not good.
- We don't understand how RSD is calculated (how much injections, how much days, etc ..). It's very important to have these informations
- We don't understand Figure 3 (SD ???), why peak areas ?
Figure 5 : why use areas and not % recovery

Additional comments

This article an interesting alternative analytical method using liquid chromatography tandem mass spectrometry (LC-MS/MS) for the determination of angiotensin peptides using nano-HPLC and PRM approaches.
Nevertheless, some important points would be changed like the use of raiodrlabelled peptides (eg AngI). and some figures would be changed. Complete absence of information from one of the main parameters when working with LC-MS(MS): matrix effect. This would be of great importance in this work, since LLE is used for sample preparation and the matrices are complex. The method seams to be validated but does not follow the criteria of an international validation guide.

Reviewer 2 ·

Basic reporting

1. Title is unclear. The authors reported the development of a novel nanoflow liquid chromatography parallel reaction monitoring mass spectrometry-based (method? platform?) for quantification of angiotensin peptides in HUVEC cultures.

2. The authors introduced the liquid chromatography-mass spectrometry methods in lines 75-86, which could contain more introduction on how this method works. The authors should also highlight the novelty of current method they are using - what are the developments the authors have done based on this existing technique as indicated in the title?

3. Abbreviations should still be defined upon first use, such as EDTA in line 142, TFA in line 147, etc.

4. Many figures do not have units for the retention time. The figures should be replotted more professionally with higher resolution and larger text. The tick labels on all the axis should also be carefully chosen. Supplementary figures are also too small to see.

Experimental design

1. Experimental methods part, line 143, 2-mercaptoethanol manufacturer and final concentration should be specified.

2. The authors detected the 7 Ang peptides using only one labeled isotope based on pre-built library. Is there any calibration steps or control experiments conducted before the experiments to verify the detection? In figure 2, the authors also mentioned the signal-to-noise ratio as “high”, could this be quantified?

3. The authors showed the sample stability for 6 hours as well as intra-day stability. The authors envision how many hours of stability this method could achieve at maximum?

Validity of the findings

1. The authors should elaborate more on the detection limit results from the built standard curves by providing an exact range in the experimental results session starting line 242.

2. The recovery rate of Ang I using SPE is significantly higher than the other Ang peptides. Could the authors elaborate more on this point?

Reviewer 3 ·

Basic reporting

no comment

Experimental design

no comment

Validity of the findings

no comment

Additional comments

The manuscript presents development of Liquid Chromatography- Paralel Reaction Monitoring Mass Spectrometry method to evaluate few angiotensin peptides in HUVEC culture supernatant. Recently various modifications of LC-MS method for angiotensins peptides analysis in cell cultures and tissues were described and successfully used in numerous studies (some examples: Hybrid Tandem Mass Spectrometry and Linear Ion Trap High-Resolution Mass Spectrometry, Matrix-Assisted Laser Desorption/Ionization Quadrupole Ion Trap Time-Of-Flight Five Tandem Mass Spectrometry, Electrospray Ionization Triple Quadrupole Mass Spectrometry etc.).
The Authors described in details their reliable work and proved high sensitivity, efficiency and precision of established method. However, there is one weakness of this work- it is limited to a few peptides only and one type of cell culture. Did the Authors try to apply described analytical technique in other type of cell culture or human/ animal tissue?

---

## Round 0.2 · accepted · Accept

The authors have satisfactorily addressed all the reviewers' concerns.

Reviewer 2 ·

Basic reporting

no comment

Experimental design

no comment

Validity of the findings

no comment

Additional comments

My comments are addressed in the revised manuscript.

Reviewer 3 ·

Basic reporting

no comments

Experimental design

no comments

Validity of the findings

no comments

Additional comments

no comments